# Chemical Gradients of Plant Substrates in an *Atta texana* Fungus Garden

Andrés Mauricio Caraballo-Rodríguez,[a] Sara P. Puckett,[b] Kathleen E. Kyle,[c] Daniel Petras,[a,d,e] Ricardo da Silva,[f] Louis-Félix Nothias,[a] Madeleine Ernst,[g] Justin J. J. van der Hooft,[h] Anupriya Tripathi,[a,i,j] Mingxun Wang,[a] Marcy J. Balunas,[b] Jonathan L. Klassen,[c] Pieter C. Dorrestein[a]

[a]Collaborative Mass Spectrometry Innovation Center, Skaggs School of Pharmacy and Pharmaceutical Sciences, University of California San Diego, La Jolla, California, USA
[b]Division of Medicinal Chemistry, Department of Pharmaceutical Sciences, University of Connecticut, Storrs, Connecticut, USA
[c]Department of Molecular and Cell Biology, University of Connecticut, Storrs, Connecticut, USA
[d]Scripps Institution of Oceanography, University of California San Diego, La Jolla, California, USA
[e]CMFI Cluster of Excellence, Interfaculty Institute of Microbiology and Medicine, University of Tuebingen, Tuebingen, Germany
[f]School of Pharmaceutical Sciences of Ribeirão Preto, University of São Paulo, Ribeirão Preto, SP, Brazil
[g]Section for Clinical Mass Spectrometry, Danish Center for Neonatal Screening, Department of Congenital Disorders, Statens Serum Institut, Copenhagen, Denmark
[h]Bioinformatics Group, Wageningen University, Wageningen, the Netherlands
[i]Division of Biological Sciences, University of California San Diego, La Jolla, California, USA
[j]Department of Pediatrics, University of California San Diego, La Jolla, California, USA

**ABSTRACT** Many ant species grow fungus gardens that predigest food as an essential step of the ants' nutrient uptake. These symbiotic fungus gardens have long been studied and feature a gradient of increasing substrate degradation from top to bottom. To further facilitate the study of fungus gardens and enable the understanding of the predigestion process in more detail than currently known, we applied recent mass spectrometry-based approaches and generated a three-dimensional (3D) molecular map of an *Atta texana* fungus garden to reveal chemical modifications as plant substrates pass through it. The metabolomics approach presented in this study can be applied to study similar processes in natural environments to compare with lab-maintained ecosystems.

**IMPORTANCE** The study of complex ecosystems requires an understanding of the chemical processes involving molecules from several sources. Some of the molecules present in fungus-growing ants' symbiotic system originate from plants. To facilitate the study of fungus gardens from a chemical perspective, we provide a molecular map of an *Atta texana* fungus garden to reveal chemical modifications as plant substrates pass through it. The metabolomics approach presented in this study can be applied to study similar processes in natural environments.

**KEYWORDS** ant fungus garden, *Atta texana*, chemical transformation, fungal symbiont, metabolomics, molecular cartography, mass spectrometry

Address correspondence to Jonathan L. Klassen, jonathan.klassen@uconn.edu, or Pieter C. Dorrestein, pdorrestein@ucsd.edu.

Here we applied mass spectrometry-based approaches to reveal chemical modifications in an ants' fungus garden. This is a step forward to future comparisons of lab-maintained and natural ecosystems.

**M**any ant species access plant-derived nutrients with the help of fungal symbionts (1). Ant fungus farming originated millions of years ago (2), likely triggered by dry and inhospitable conditions (3). Leaf-cutter ants grow a specific cultivar fungus in specialized underground structures called fungus gardens as their main food source (4, 5). This cultivar fungus breaks down forage material such as leaves provided by the ants to obtain the necessary nutrients for its own growth (6). In turn, the ants eat the fungus' specialized hyphal tips, known as gongylidia, which contain degradative enzymes from the cultivar (7) and nutrients that are metabolically available to the ants (8). Many of these enzymes pass through the ant gut intact and then are spread

throughout fungus gardens by the ants to facilitate the digestion of newly incorporated substrates (9–11). These complementary roles of ants and their fungal symbiont have been demonstrated by monitoring carbon and nitrogen sources in the diet, indicating that the fungal symbionts partially meet ant nutritional needs (12), perhaps with some assistance from the fungus garden microbiome (13–15). Fungal enzymes present in the garden are responsible for plant biomass degradation (6, 16–18), especially transforming plant metabolites such as polysaccharides and phenolic compounds (19–22). Fungus gardens from leaf-cutter ants have been compared to bioreactors (21) and human compost (23) and have even been described as external ant guts (24) due to their capacity to process plant constituents. Few studies have directly assayed small molecules from fungus gardens (24–26), though some have shown the differential distribution of fungal metabolic enzymes in different regions of the fungus garden (25, 27). That distribution of enzymes reflects the incorporation of fresh plant material at the top of leaf-cutting ant fungus gardens followed by its sequential degradation while moving through the garden, with recalcitrant biomass being removed by ants from the bottom of the fungus garden as trash (28, 29). Nonetheless, maps of small molecule diversity in ant fungus gardens have remained unavailable due to the lack of computational workflows that go beyond the analysis of a few selected metabolites, which did not exist until recently.

Here, we highlighted chemical transformations in a laboratory-maintained *Atta texana* fungus garden using molecular networking (30–32), three-dimensional (3D) cartography (33), and mass shift analysis (34). The use of nontargeted metabolomics, via liquid chromatography-tandem mass spectrometry (LC-MS/MS) (32, 35–37), and *in silico* annotation and classification of detected metabolites (32, 38) enabled us to identify metabolite features that chemically differentiate fungus garden regions as plant substrates pass through. Furthermore, we identified the types of chemical transformations that are carried out based on the differential abundance of compounds that occur among the sampled layers of the fungus garden. The observed transformations provide insight into the chemistry and the modification of molecules that result from potential chemical transformations or differential degradation inside an ant fungus garden.

## RESULTS AND DISCUSSION

**Molecular cartography in the fungus garden.** A colony of *Atta texana* fungus-growing ants was maintained in the laboratory environment and provisioned with maple leaves, which the ants cut and incorporated into their fungus garden to predigest these materials. Following this predigestion by the cultivar fungus, recalcitrant plant biomass became trash that the ants removed from the garden (Fig. 1). Our metabolomics approach enabled us to chemically differentiate regions of the fungus garden (see Fig. S1 to S3 in the supplemental material) and visualize the distribution of detected molecules, such as ergosterol peroxide (compound 1) (Fig. 1), ginnalin A (compound 2) (Fig. 2), (*E*)-9-oxo-11-(3-pentyloxiran-2-yl)undec-10-enoic acid (compound 3) (Fig. 2), phytosphingosine (compound 4) (Fig. 2), and other representative members from the molecular families described below (see Fig. S4 to S9). The relative abundance of molecules from the maple leaves, such as saccharide-decorated flavonoids and phenolic compounds (Fig. S4 to S6), decreased when moving from the top to the bottom of the fungus garden, in contrast to other compounds that increased in relative abundance across these layers (Fig. S2). These gradients are due to either chemical modifications or preferential degradation of less abundant compounds (Fig. S4 to S9). We observed that fungus garden and trash samples were enriched with phytosphingosines (Fig. 2 and S7), amino alcohols that are also present in plants (39), whereas features (metabolites) associated with the trash material were enriched in steroids, such as the fungal metabolite ergosterol peroxide (compound 1) (40, 41), (Fig. 1 and S7), and features $m/z$ 498.3932 and $m/z$ 453.3341 that are consistent with molecular formulas of $C_{32}H_{52}NO_3$ (calculated [calc.] $m/z$ 498.3941; error, 1.9 ppm) and $C_{30}H_{45}O_3$

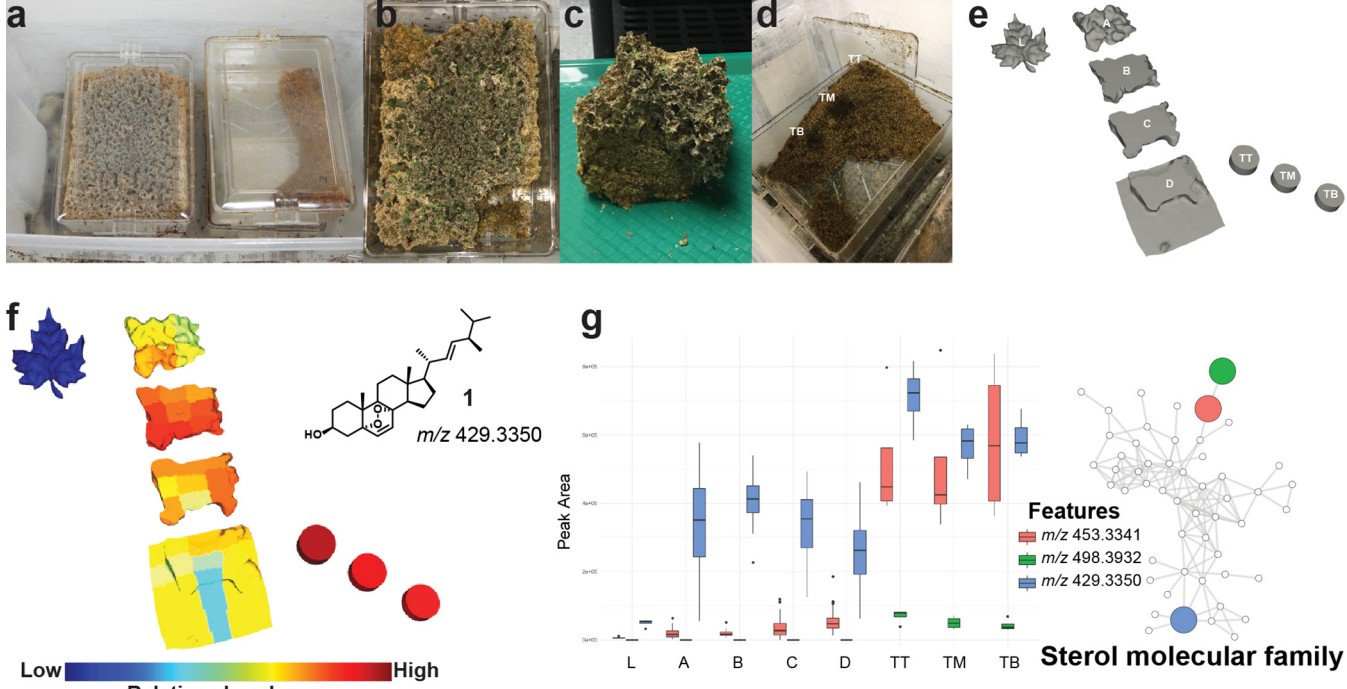

**FIG 1** Deconstruction and molecular signatures from *A. texana* fungus garden. (a) Plastic containers containing an *Atta texana* fungus garden (left) and trash material removed by ants (right). Ants have free access to both chambers. (b) Location of the removed portion of the fungus garden used for mass spectrometry analysis (lower right corner), shown in panel c. (c) The green fragments at the top of the fungus garden are freshly incorporated maple leaves. (d) Chamber containing the trash material removed from the fungus garden by the ants. Three sampling locations from the trash are highlighted as "TB," "TM," and "TT." (e) Deconstruction of the fungus garden portion shown in panel c that was used for mapping detected molecules, including maple leaves (left side, labeled as "L") placed in the outer colony box that ants cut and incorporated into the top of the fungus garden, layers of the fungus garden from top to bottom (A, B, C, and D), and, at the right side of the figure, the three sampled locations from the trash chamber, from top to bottom (TT, TM, and TB). (f) Distribution of ergosterol peroxide (compound 1) in the deconstructed fungus garden, detected as *m/z* 429.3350. (g) Abundances of features associated with trash material belonging to the sterol molecular family (ergosterol peroxide [compound 1], feature *m/z* 498.3932 and feature *m/z* 453.3341). Boxes represent the 25%, 50%, and 75% quartiles, and the whiskers extend ±1.5 times the interquartile range. The molecular cartography of this deconstructed *Atta texana* fungus garden visualized in 'ili (33) is shown in the online video available at https://youtu.be/_ikhKelfrY8 (see "*Atta texana* fungus garden deconstruction" in Materials and Methods for details).

(calc. *m/z* 453.3363; error, 4.9 ppm) and belong to the same molecular family (Fig. 1). Common and highly reactive plant metabolites known as oxylipins were also detected in the fungus garden and trash material (Fig. S8). Oxylipins, such as compound 3, are signaling molecules that originate from the oxidation of polyunsaturated fatty acids and that are involved in plant defense (42, 43). Their presence is consistent with oxidation processes occurring in the fungal garden, where the oxylipins provide carbon sources that are available to the fungi.

Figure 2 shows gradients of other plant-derived metabolite abundances from the top to the bottom of the fungus garden. These gradients were also observed for triterpenoid derivatives (Fig. S9), leading to the highest abundances of these compounds at the bottom of the fungus garden and in the trash. These gradients parallel the metabolic transformations of food components in the digestive tracts of animals, such as those involving the metabolism of flavonoids, steroids (molecules with steroidal cores), and fatty acids (44–46). These gradients are consistent with known gradients of polysaccharide degradation in leaf-cutting ant fungus gardens, where fresh leaves are incorporated at the top of the garden and then sequentially digested before recalcitrant material is removed from the garden as trash (7, 25, 27, 28). Maple leaves, fungus garden layers, and trash were each chemically distinct, consistent with their distinct roles in the system as the substrate, the site of active predigestion, and spent waste, respectively (Fig. S3).

**Chemical modifications in the fungus garden.** Digestive processes generate modified products whose precursors are consumed. To provide an overview of putative

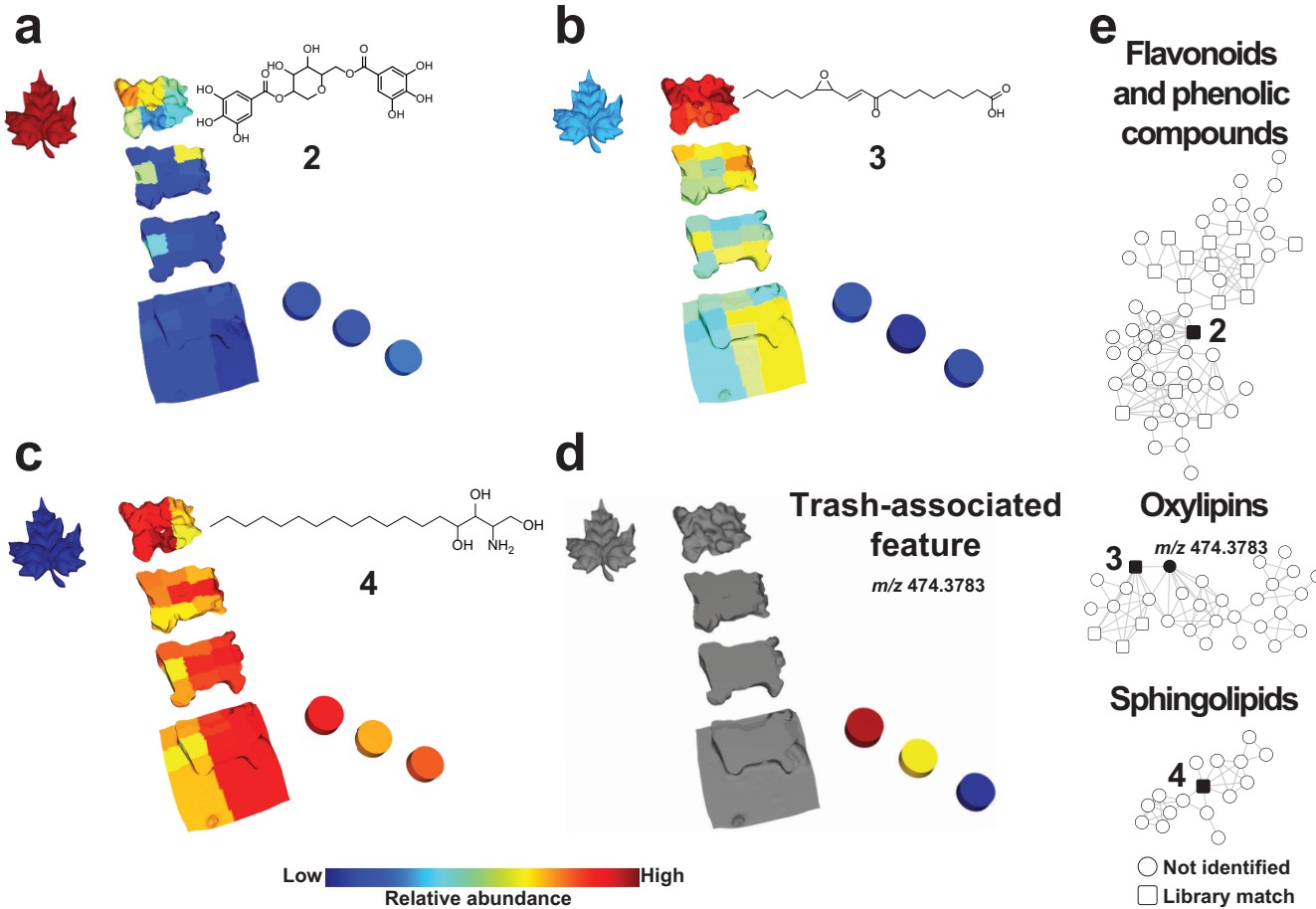

**FIG 2** Spatial distributions of molecular signatures from *A. texana* fungus garden. The spatial distributions of ginnalin A (compound 2), detected as *m/z* 469.0971 (a), (*E*)-9-oxo-11-(3-pentyloxiran-2-yl)undec-10-enoic acid [*trans*-EKODE-(E)-lb] (compound 3), detected as *m/z* 311.221 (b), phytosphingosine (compound 4), detected as *m/z* 318.2995 (c), and an unknown feature associated with the trash material, detected as *m/z* 474.3783 (d). (e) Molecular families of flavonoid and phenolic compounds, oxylipins, and sphingolipids, highlighting in black the molecules visualized in panels a to d. Connected nodes in the networks correspond to structurally related molecules based on their spectral similarity.

metabolic transformations occurring in ants' fungus gardens, we combined mass shift analysis (34) and calculated the relative metabolite abundances for metabolite pairs from each fungus garden section by using a proportionality score (see Materials and Methods) (47). By considering the proportions between the relative abundances of two chemically related molecules (i.e., connected nodes in a network), their mass shifts and the modifications that these imply (e.g., a 15.996-Da shift indicates a gain or loss of oxygen, and a 2.015-Da shift indicates an oxidation or reduction via the loss or addition of $H_2$), and their distribution between two locations (maple leaves, layers of fungus garden, and layers of trash material), we can discover related molecules that have the largest differential abundance between the compared locations (see Materials and Methods) (Fig. 1 and 3; Fig. S10). It should be noted that this approach cannot differentiate between different types of changes in the absolute abundance of each molecule, e.g., chemical transformation leading to the accumulation of a molecule or the complete degradation of a molecule leading to its decreased abundance. However, the chemical similarities and relative abundances between each pair of molecules that are identified using molecular networking imply relationships between these molecules that are consistent with each molecule belonging to the same molecular family (48). We interpret abundance changes that occur across layers to be largely driven by anabolic or catabolic pathways, potentially linked to enzymatically mediated transformations. Absolute molecular concentrations might also be altered by additions from the external environment, but the closed nature of the laboratory-maintained ant fungus

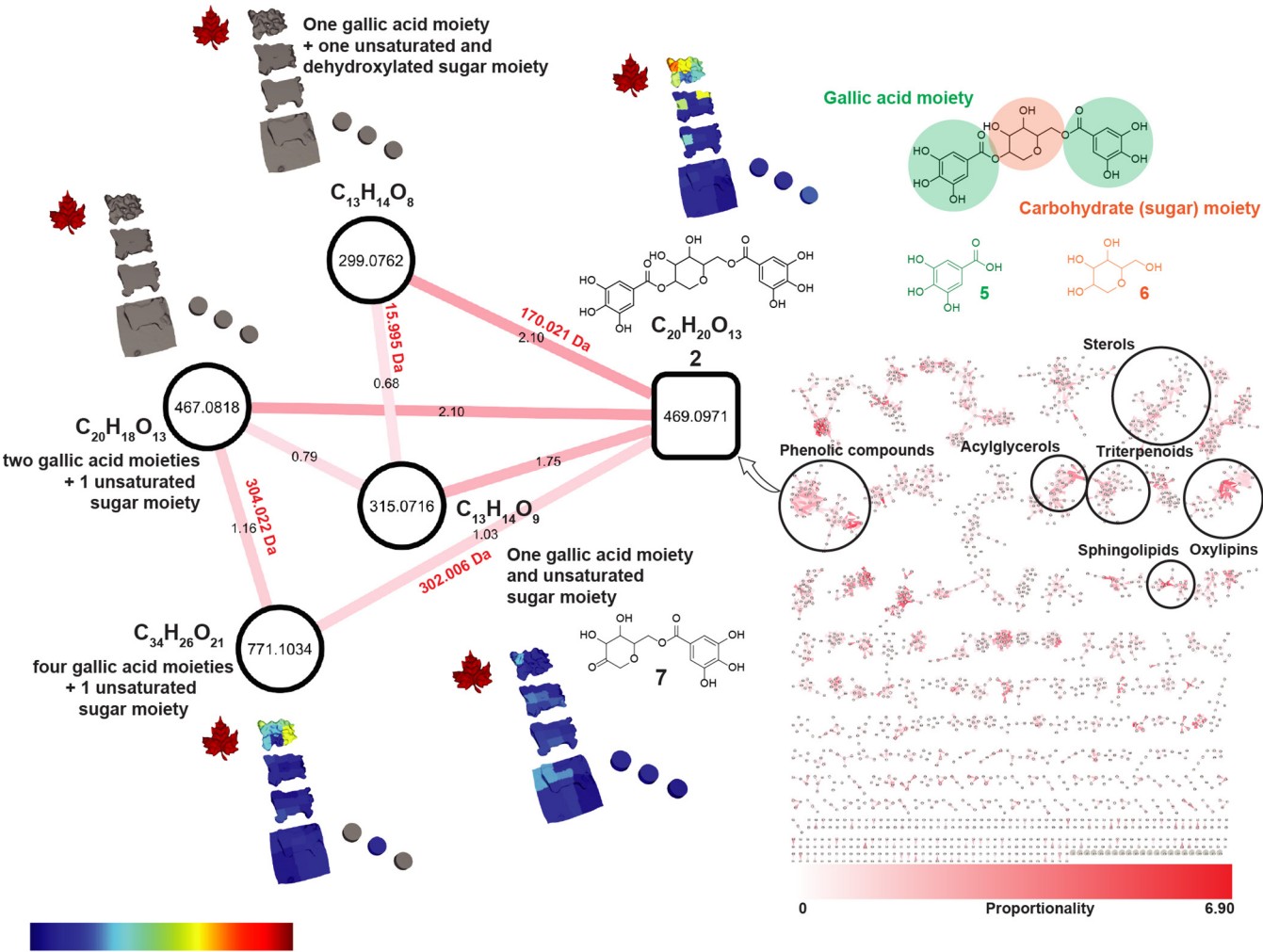

**FIG 3** Potential transformations of phenolic compounds related to ginnalin A. Chemical features are highlighted from a molecular network (right) based on their high proportionality scores (red edges, edge labels are in black). The spatial distributions of the highlighted features chemically related to compound 2 by the proportionality approach are also shown in the 3D model of the *Atta texana* fungus garden (relative abundances are shown following the color code in the figure). Chemical structures of representative candidates from this molecular family (compounds 2 and 7) and substructures (compounds 5 and 6) are shown.

gardens means that such additions essentially only occur directionally via the leaves when they are incorporated into the fungus garden by the ants.

In this study, we applied the chemical proportionality approach (47) by modifying the meta-mass shift analysis (34) to also consider the relative abundances of the detected molecules, which quickly highlights features that are potentially involved in chemical transformations. Proportionality scores highlighted such changing pairs of nodes occurring in various locations in the *A. texana* fungus garden, and the associated mass shifts provided evidence of the types of modifications occurring in the sampled locations. An example of this proportionality approach presented in Fig. 3 shows the feature corresponding to *m/z* 469.0971 (calc. *m/z* 469.0976; error, 1.2 ppm), annotated as the bioactive phenolic compound ginnalin A (compound 2) (49), to be involved in chemical transformations. Based on the information regarding mass shifts and the similarity of fragmentation spectra (MS/MS), structural modifications involving compound 2 were suggested. All the fragmentation spectra from the nodes that are directly connected to compound 2 in the molecular network shared the *m/z* 153.02 base peak corresponding to a phenolic substructure (gallic acid moiety, compound 5 in Fig. 3), indicating that a putative double bond is located in the sugar moiety (compound 6 in Fig. 3). Compound 2 was detected in the leaves and in the fungus garden, while the

features corresponding to m/z 299.0762 (calc. m/z 299.0761; error, −0.2 ppm) and m/z 467.0818 (calc. m/z 467.0972; error, −0.7 ppm) were only detected in the plant material, possibly indicating both molecules are transformed in the fungus garden to either compound 2 or a related molecule, such as features m/z 315.0717 (calc. m/z 315.0710; error, −1.7 ppm; compound 7) and m/z 771.1034 (calc. m/z 771.1039; error, 0.7 ppm). Modification of phenolic compounds in fungus gardens has been described as a mechanism of detoxifying these common plant defenses against defoliating herbivores (6, 20, 50, 51). Additionally, gallic acid (compound 5) can be produced by endophytic microorganisms (52), suggesting another interesting source of these phenolic derivatives.

Proportionality scoring is a logarithmic expression, and we selected an absolute value of 1 as prioritization cutoff, because a score close to zero indicates a low ratio of differing abundance between two chemically related molecules in two sample locations resulting from a small change in abundances between the two features (see the definition and calculation of the proportionality scores in Materials and Methods). Differences representing a gain or loss of $H_2$ (2.015 Da) were the predominant type of chemical transformation observed throughout the data set, being one of the most frequent mass shifts with a proportionality score of >1 among leaves, the fungus garden, and trash layers (Fig. 4). This common modification was observed in molecular families that contain phenolic compounds (Fig. 3) and phytosphingosines (Fig. S7). Mass shifts corresponding to $CH_2$ (14.015 Da) and $C_2H_4$ (28.031 Da) were other common changes observed in the top layer of the fungus garden and between the bottom layer of the fungus garden and the trash (Fig. 4). Oxidation or dehydroxylation combined with reduction results in the gain or loss of oxygen that can be detected as a mass difference of 15.995 Da. The transformations corresponding to these differences were more frequently observed at the top layer of the fungus garden during the breakdown of flavonoids and phenolic compounds (Fig. 4; Fig. S4 to S6). Chemical transformations consistent with the addition or removal of sugar moieties in the A. texana fungus garden were also highlighted by proportionality scores of >1. These transformations, corresponding to mass differences of 162.053 Da ($C_6H_{10}O_5$), were associated with plant material and the top layers of the fungus garden (Fig. 4) and involved plant metabolites belonging to the acylglycerol molecular family (compounds 8 and 9) (Fig. 5 and Fig. S10) and flavonoids such as quercetin-3-O-pentoside (compound 10) and quercetin (compound 11).

The existence of chemical gradients in the fungus garden resembles a predigestion process where substrates are modified to facilitate their consumption by the ants and residues are generated for removal as trash, as exemplified here by plant constituents that pass through an ant fungus garden ecosystem. Our study enabled us to detect molecular families and chemical transformations occurring in a fungus garden. These results will enable further investigations concerning the roles of these metabolites, as has been recently demonstrated for fatty acids in fungus gardens (24) and for plant volatile compounds that are modified by fungus garden-associated bacteria (26). Because previous reports have shown that the enzymes produced in the fungus garden vary depending on the available substrates (53), we expect that the chemical modifications and the types of transformations observed in this study will also vary based on changes in the substrates available for the colony. Additionally, environmental factors such as temperature or humidity and the composition of microbiomes that are associated with ants and their fungus gardens (26) will likely also influence these modifications in natural ecosystems.

In summary, the 3D cartographic analyses performed in this study provide an overview of chemical changes occurring in an A. texana fungus garden. Complementary to other studies exploring the capacity of fungus gardens to metabolize plant substrates (22, 24), our results also show that specific chemical transformations of plant components are associated with certain regions of the fungus garden and that the number of modifications are more extensive than previously described (24, 53). These results

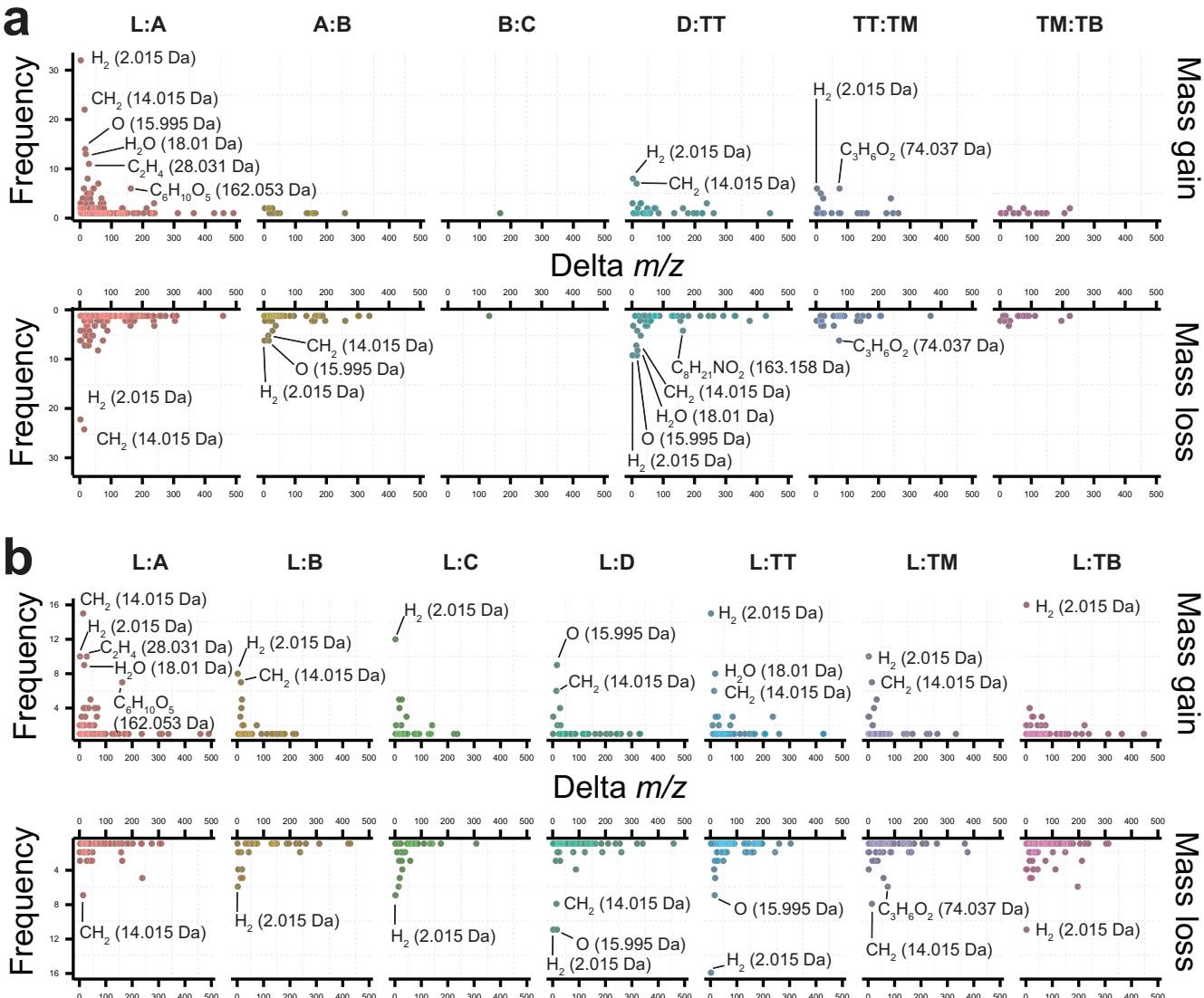

**FIG 4** Frequency of delta masses observed in metabolomics data from a deconstructed *Atta texana* fungus garden. Proportionality metrics were calculated between compounds found in different layers of the fungus garden and between leaves, fungus garden, and trash material. (a) Frequency of delta masses calculated between samples corresponding to leaves (L), layers of fungus garden from top to bottom (A, B, C, and D), and trash material (samples collected from the chamber that ants use to deposit the trash material, from the top [TT] middle [TM], and bottom [TB]). (b) Frequency of delta masses between compounds found in leaves (L) and each of the fungus garden and trash layers. Mass shifts between network pairs with proportionality scores of >1 indicated chemical transformations prevalent in specific locations. Frequencies of annotated mass shifts observed from network pairs with proportionality values of >1 indicated that most transformations occurred between leaves and the top layer of fungus garden (layer A), between the bottom of the fungus garden (layer D) and the trash, and between the trash layers (TT, TM, and TB). None of the observed mass shifts resulted in a high proportionality score calculated between fungus layers C and D (proportionality C:D). The figures in panel b shows the most frequent mass shifts observed for network pairs with the highest transformation rates calculated from proportionality between leaves (L) and each sample group. This indicates that most of these transformations are observed throughout the fungus garden but also that some are specific to the trash, such as $C_8H_{21}NO_2$ (163.158 Da) (as observed in panel a).

agree with the metabolic role of fungus gardens, where fungal predigestion of biomass complements ant-associated enzymes (20, 54). We also captured the known gradients in the metabolism of plant-derived molecules from top to bottom of the fungus garden (25), where plant material is predigested starting when leaves enter the fungus garden and continuing through to the bottom layer of the fungus garden, after which some recalcitrant molecules are removed from the system as trash (Fig. 5). We provided a way to retrieve chemical information that will help understand these predigestion processes and also to potentially identify chemical cues associated with ant behaviors, such as how the ant chooses unwanted fungal garden material to discard. Capturing these chemical transformations in detail is a fundamental step to understanding the influence of molecules in

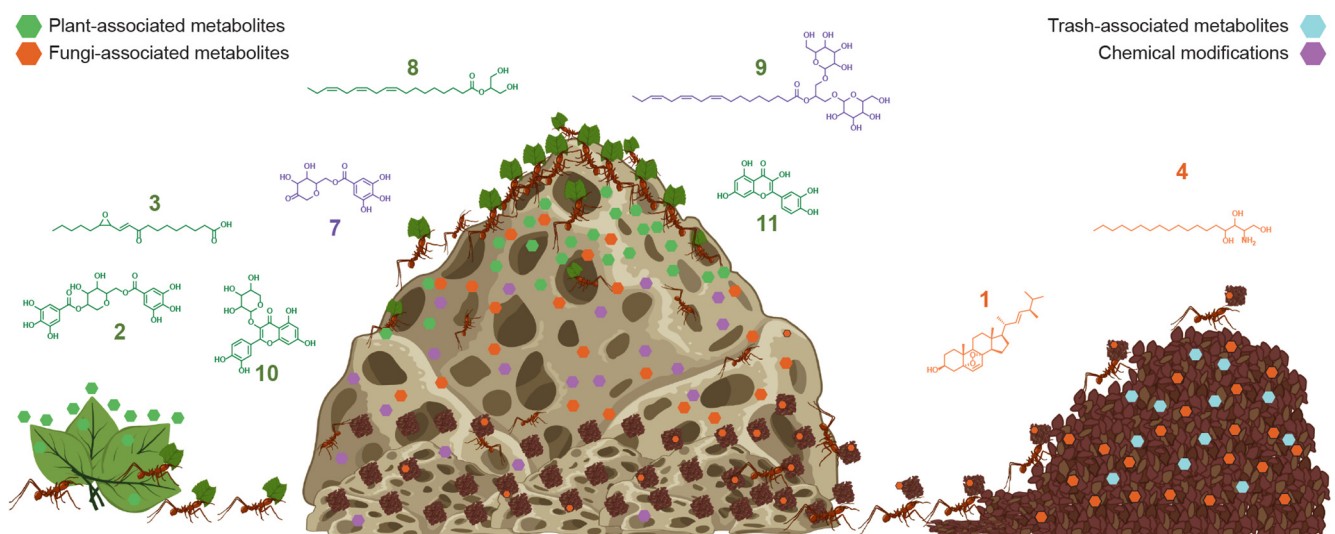

**FIG 5** Summary of detected molecules and chemical modifications detected in an *Atta texana* fungus garden. Schematic representation of a laboratory-maintained *Atta texana* fungus garden. From left to right, leaves carried by ants to the fungus garden (middle) and trash accumulated by ants (right). The chemical gradient of plant metabolites showing high abundance at the top layer of the fungus garden is consistent with chemical modifications occurring across the fungus garden. High abundance of fungal metabolites associated with the trash indicates that these are chemical labels of the discarded material removed by the ants. Some examples of detected molecules and chemical modifications are shown. Chemical structures of trash-associated metabolites remain to be elucidated. Color codes of the compounds are as indicated at the top of the figure. This image was partially created using BioRender.

fungus gardens on the microbial communities inhabiting these systems. Our methodologies provide a very detailed overview of metabolic processes occurring in a laboratory-maintained *A. texana* fungus garden, and we expect the approach can be leveraged to unravel similar processes in natural environments to compare between natural and lab-maintained ecosystems.

## MATERIALS AND METHODS

**General overview of workflows applied in this study.** To analyze chemical transformations occurring in leaf-cutting ant fungus gardens, we deconstructed a laboratory-maintained *Atta texana* fungus garden (Fig. 1). We applied spectral alignment using molecular networking (30, 31), analyzed the spatial distribution of molecular features using molecular cartography (33), and annotated MS/MS spectra via Global Natural Products Social Molecular Networking (GNPS) workflows, including MolNetEnhancer (31, 32, 36), to identify molecular signatures from the fungus garden samples. The false-discovery rate (FDR) for the compound annotations was estimated using the Passatutto decoy-based method (55). We further applied the chemical proportionality (proportionality) approach (47) using a modification of the recently introduced meta-mass shift analysis (34), by also considering the abundances of the detected molecules to quickly highlight the features (metabolites) potentially involved in chemical transformations. Samples were prepared for untargeted profiling via reverse-phase liquid chromatography-tandem mass spectrometry (LC-MS/MS). Data were collected using data-dependent acquisition and by fragmenting the five most abundant precursor ions. Feature detection was performed using the open source software MZmine 2 (56, 57). Spectral alignments of the acquired MS/MS spectra and 3D visualization were performed using the Global Natural Products Social Molecular Networking (GNPS) platform (30, 31, 33). By combining these approaches, we obtained molecular signatures from the fungus garden that enabled us to reveal chemical transformations occurring in specific locations of this system.

***Atta texana*** **fungus garden.** A queenright *A. texana* colony with brood was maintained in the laboratory of Jonathan Klassen in the department of Molecular and Cell Biology, University of Connecticut, for 2 years before the deconstruction conducted for this study according to standard methods for maintaining living colonies in the laboratory (58). Colony JKH000189 was collected from Clear Creek Wildlife Management Area, LA (31.04990, −93.40245), under Louisiana Department of Wildlife and Fisheries permit WL-Research-2016-10. Colony JKH000189 was maintained in plastic chambers where ants accessed flash-frozen maple leaves (primarily *Acer platanoides*, but also *Acer rubrum* and *Acer saccharum*) provided every 3 days. An empty plastic chamber was also provided that ants used to contain their waste (Fig. 1).

***Atta texana*** **fungus garden deconstruction.** A 10- by 10- by 10-cm piece of fungus garden was removed from the chamber and scanned for creation of a virtual 3D image using a Structure Sensor Mark II and the Structure app (Occipital, Inc., Boulder, CO). The virtual 3D model was created using Meshmixer software (Autodesk Inc., San Rafael, CA), and the coordinates for 'ili visualization were obtained using MeshLab software (http://www.meshlab.net). The piece of fungus garden was then sliced into four layers, and each layer was further divided into nine pieces of approximately 3 by 3 by 3 cm (or 27 cm³) and homogenized by

**TABLE 1** Reference standards used to confirm annotated members of molecular families detected from *Atta texana* fungus garden

| | *m/z* [M + H]$^+$ | | Error (ppm) | Molecular formula | Accurate mass (Da) | CAS no. | Retention time (min) |
|---|---|---|---|---|---|---|---|
| Reference compound | Detected | Calculated | | | | | |
| Ergosterol peroxide (compound 1) | 429.3355 | 429.3363 | 1.9 | $C_{28}H_{44}O_3$ | 428.3285 | 2061-64-5 | 8.13 |
| EKODE (*E*)-9-oxo-11-(3-pentyloxiran-2-yl)undec-10-enoic acid (compound 3) | 311.2212 | 311.2217 | 1.6 | $C_{18}H_{30}O_4$ | 310.2139 | 478931-82-7 | 5.83 |
| Phytosphingosine (compound 4) | 318.2995 | 318.3003 | 2.4 | $C_{18}H_{39}NO_3$ | 317.2924 | 554-62-1 | 5.52 |
| Kaempferol$^a$ | 287.0545 | 287.0550 | 1.8 | $C_{15}H_{10}O_6$ | 286.0472 | 520-18-3 | 3.55 |
| Quercetin$^b$ (compound 9) | 303.0498 | 303.0499 | 0.4 | $C_{15}H_{10}O_7$ | 302.0421 | 117-39-5 | 3.35 |

$^a$See Fig. S4 in the supplemental material.
$^b$See Fig. S5.

vortexing. Approximately 100 mg of each sample was extracted three times with 2:1 dichloromethane (DCM)-methanol (MeOH), sonicated for 10 min, dried under a stream of gaseous nitrogen, and shipped to the Dorrestein Laboratory at UC San Diego for LC-MS/MS acquisition. A short video showing the spatial distribution of the detected molecules from the deconstructed *Atta texana* fungus garden can be accessed at https://youtu.be/_ikhKelfrY8. Briefly, using the table of feature abundances containing the LC-MS/MS data (see "Feature-based molecular networking" described below) and the virtual 3D model, we show the distribution of each molecular signature detected in the fungus garden using 'ili for visualization (https://ili.embl.de/).

**Annotation of detected features.** The automatic annotation of detected molecules via GNPS workflows (31, 32, 36) was manually confirmed for selected features. Representative compounds from the main chemical classes discussed in the main text were confirmed by using reference standards. This confirmation corresponds to level 1 annotation, while annotation by spectral match and at the molecular family correspond to levels 2 and 3 according to the 2007 metabolomics initiative (59). The following standards were used to confirm annotations at level 1 using spectral matches, accurate masses, and retention times: ergosterol peroxide (compound 1) (Carbosynth LLC), (10*E*)-9-oxo-11-(3-pentyl-2-oxiranyl)-10-undecenoic acid (compound 3) (Cayman Chemicals Company, Inc.), phytosphingosine (compound 4) (Sigma-Aldrich), kaempferol (see Fig. S4 in the supplemental material) (VWR International, LLC) and, quercetin (compound 9) (Fig. S5) (VWR International, LLC). See Table 1.

**LC-MS/MS conditions.** Samples were resuspended in 100% methanol containing 2 $\mu$M sulfamethazine as an internal standard, and LC-MS/MS analysis was performed in an UltiMate 3000 ultraperformance liquid chromatography (UPLC) system (Thermo Scientific) using a Kinetex 1.7-$\mu$m C$_{18}$ reversed-phase ultrahigh-performance liquid chromatography (UHPLC) column (50 by 2.1 mm) and Maxis Q-time of flight (TOF) mass spectrometer (Bruker Daltonics) equipped with an electrospray ionization (ESI) source. The column was equilibrated with 5% of solvent B (LC-MS-grade acetonitrile, 0.1% formic acid) for 1 min, followed by a linear gradient from 5% to 100% of solvent B over 8 min, and then held at 100% solvent B for 2 min. Then, the gradient was reduced from 100% to 5% of solvent B over 0.5 min and then maintained at 5% solvent B for 2.5 min, using a flow rate of 0.5 ml/min throughout the run. MS spectra were acquired in positive ion mode in the range of 100 to 2,000 *m/z*. A mixture of 10 mg/ml each of sulfamethazine, sulfamethizole, sulfachloropyridazine, sulfadimethoxine, amitriptyline, and coumarin was run after every 96 injections for quality control. An external calibration with ESI-low concentration tuning mix (*m/z* 118.086255, 322.048121, 622.028960, 922.009798, 1221.990637, 1521.971475, and 1821.952313) (Agilent technologies) was performed prior to data collection. An internal calibrant hexakis(1*H*,1*H*,2*H*-perfluoroethoxy) phosphazene (CAS 186817-57-2) was used throughout the runs. A capillary voltage of 4,500 V, nebulizer gas pressure (nitrogen) of 2 bar, ion source temperature of 200°C, dry gas flow of 9 liters/min source temperature, spectral rate of 3 Hz for MS1 and 10 Hz for MS2 was used during each run. For acquiring MS/MS fragmentation, the 5 most intense ions per MS1 were selected, the MS/MS active exclusion parameter was enabled, set to 2 and to release after 30 s, and the precursor ion was reconsidered for MS/MS if the current intensity/previous intensity ratio was >2. An advanced stepping function was used to fragment ions according to the settings in Tables 2 and 3.

**Feature-based molecular networking.** Feature finding was performed with the open source MZmine software (56) version 2.38 using the settings shown in Table 4. These preprocessing steps (Table 4) generated the .mgf and quantification tables used in the GNPS feature-based molecular network workflow.

**False-discovery rate: Passatutto.** The Passatutto decoy-based method 43 was used to estimate an FDR for the annotations with the library match settings used for spectral identification. An FDR pf <0.072 was obtained for the annotations using a minimum matching fragment ions of 6 (GNPS job link [Passatutto], https://gnps.ucsd.edu/ProteoSAFe/status.jsp?task=c7a1733750bd415fa32176e125fac42e). An FDR of <0.16 was obtained for the obtained annotations using a minimum matching fragment ions of 5 (GNPS job link [Passatutto] https://gnps.ucsd.edu/ProteoSAFe/status.jsp?task=206023ccdbd747ee885aab2b16dfcf31).

**Feature-based molecular network for deconstruction of *A. texana* fungus garden.** A molecular network was created with the feature based molecular networking workflow (https://ccms-ucsd.github.io/GNPSDocumentation/featurebasedmolecularnetworking/) on the GNPS website (http://gnps.ucsd.edu). The data were filtered by removing all MS/MS fragment ions within ±17 Da of the precursor *m/z*. MS/MS spectra were window filtered by choosing only the top 6 fragment ions in the ±50-Da window throughout the spectrum. The precursor ion mass tolerance was set to 0.02 Da and a MS/MS fragment ion tolerance of 0.02 Da. A network was then created where edges were filtered to have a cosine score

**TABLE 2** Instrument settings for data-dependent acquisition of *Atta texana* fungus garden samples

| Time[a] (%) | Collision RF[b] (Vpp) | Transfer time ($\mu$s) | Collision value |
|---|---|---|---|
| 0 | 450.0 | 70.0 | 125 |
| 25 | 550.0 | 75.0 | 100 |
| 50 | 800.0 | 90.0 | 100 |
| 75 | 1100.0 | 95.0 | 75 |

[a]Collision stepping switch time.
[b]RF, radio frequency; Vpp, volts peak to peak.

of >0.7 and more than 6 matched fragment ions. Furthermore, edges between two nodes were kept in the network if and only if each of the nodes appeared in each other's respective top 10 most similar nodes. Finally, the maximum size of a molecular family was set to 100, and the lowest scoring edges were removed from molecular families until the molecular family size was below this threshold. The spectra in the network were then searched against GNPS' spectral libraries. The library spectra were filtered in the same manner as the input data. All matches kept between network spectra and library spectra were required to have a score of >0.7 and at least 6 matched fragment ions. Molecular networks were visualized using Cytoscape (60), version 3.7.2. (GNPS FBMN job link, https://gnps.ucsd.edu/ProteoSAFe/status.jsp?task=5df1dc83e075478ba69d1bb41bf9499a).

**In silico annotation of detected features from *A. texana* fungus garden using GNPS workflows.** Network annotation propagation (NAP) (36) was performed via the GNPS platform. The parameters used the 10 first candidates for consensus score, 15 ppm accuracy for exact mass candidate search, positive acquisition mode, and 0.5 cosine value to subselect inside a cluster. Fusion results were used to determine a consensus, searching only for [M+H]$^+$ adduct type. A maximum of 10 candidate structures were used in the graph. The following databases were searched: Dictionary of Natural Products (DNP), Super Natural II (61), GNPS, and Chemical Entities of Biological Interest (ChEBI) (62) (GNPS NAP job link, https://proteomics2.ucsd.edu/ProteoSAFe/status.jsp?task=05f63922be7244bdb9aa42784be0a6eb). Automatic workflows for peptide analogues was performed using VarQuest (63) in GNPS and can be accessed at https://gnps.ucsd.edu/ProteoSAFe/status.jsp?task=cba438bedbb04f73afd4bd7a85ef665d.

A complementary analysis for substructures present in the data set was performed using the mass-to-motif MS2LDA (37) workflow in the GNPS and can be accessed at https://gnps.ucsd.edu/ProteoSAFe/status.jsp?task=b3bb2654ac9542b7a8a8fcf86001e215.

**MolNetEnhancer for molecular network of deconstructed *A. texana* fungus garden.** Molecular network annotations were enhanced via the MolNetEnhancer workflow (32) merged in the GNPS platform. The workflow merged *in silico* annotations from the network annotation propagation (NAP) (36), VarQuest (63), and MS2LDA (37) to provide structures annotations at the class level (MolNetEnhancer job link, https://gnps.ucsd.edu/ProteoSAFe/status.jsp?task=50dfaf589c8140bd85a8ca199db59de3).

**Proportionality score.** The proportionality score was calculated between two directly connected nodes across the entire molecular network using the following equation:

$$\text{proportionality} = \log \frac{N_{s1}/M_{s1}}{N_{s2}/M_{s2}},$$

where $N_{s1}$ and $M_{s1}$ correspond to the peak area of the detected features $N$ and $M$ in sample S1, while $N_{s2}$ and $M_{s2}$ correspond to the peak area of the detected features $N$ and $M$ in sample S2. A constant ($k = 1.0$ e$^{-10}$) is added to each value to avoid including any zero values during the calculation. Chemical proportionality scores are available within the GNPS environment and can be accessed through the following links: https://proteomics2.ucsd.edu/ProteoSAFe/status.jsp?task=059d652858134586a5320dc94e2d732e (chemical proportionality table calculated between each sample type: L:A, A:B, B:C, C:D, D:TT, TT:TM, and TM:TB) and https://proteomics2.ucsd.edu/ProteoSAFe/status.jsp?task=da94680c01594122b7ef88a1070155ff (chemical

**TABLE 3** CID energies for MS/MS data acquisition used in this study[a]

| Type | Mass (Da) | Width | Collision value | Charge state |
|---|---|---|---|---|
| Base | 100.00 | 4.00 | 22.00 | 1 |
| Base | 100.00 | 4.00 | 18.00 | 2 |
| Base | 300.00 | 5.00 | 27.00 | 1 |
| Base | 300.00 | 5.00 | 22.00 | 2 |
| Base | 500.00 | 6.00 | 35.00 | 1 |
| Base | 500.00 | 6.00 | 30.00 | 2 |
| Base | 1,000.00 | 8.00 | 45.00 | 1 |
| Base | 1,000.00 | 8.00 | 35.00 | 2 |
| Base | 2,000.00 | 10.00 | 50.00 | 1 |
| Base | 2,000.00 | 10.00 | 50.00 | 2 |

[a]CID, collision-induced dissociation. The mass of internal calibrant was excluded from the MS/MS list using a mass range of *m/z* 621.5 to 623.0.

**TABLE 4** Preprocessing settings for feature detection using MZmine 2 of LC-MS/MS acquired data from *Atta texana* fungus garden samples

| Setting[a] | Value |
|---|---|
| **Mass detection** | |
| MS1 | 1.0E4 |
| MS2 | 1.0E2 |
| **Chromatogram building** | |
| Min. time span (min) | 0.01 |
| Min. ht | 3.0E4 |
| Tolerance (ppm) | 25 |
| **Deconvolution (baseline cutoff algorithm)** | |
| Min. peak ht | 1.0E4 |
| Peak duration range (min) | 0.01–1.0 |
| Baseline level | 1.0E2 |
| *m/z* range for MS2 scan pairing (Da) | 0.01 |
| RT range for MS2 scan pairing (min) | 0.3 |
| **Isotopic peak grouper** | |
| *m/z* tolerance (ppm) | 25 |
| RT tolerance (min) | 0.2 |
| Max. charge | 2 |
| **Alignment (join aligner)** | |
| *m/z* tolerance (ppm) | 25 |
| wt for *m/z* (%) | 75 |
| wt for RT (%) | 25 |
| RT tolerance (min) | 0.2 |
| RT correction | Checked |
| **Gap filling (peak finder)** | |
| Intensity tolerance (%) | 1 |
| *m/z* tolerance (ppm) | 25 |
| RT tolerance (min) | 0.2 |
| RT correction | checked |
| **Peak filter** | |
| Peak area | 1.0E4–1.0E7 |
| **Peak row filtering to export .mgf file to GNPS** | |
| Min. peaks in a row (no.) | 2 |
| RT (min) | 1.00–14.00 |

[a]Min., minimum; RT, retention time; Max., maximum.

proportionality table calculated between leaves and the entire data set: L:A, L:B, L:C, L:D, L:TT, L:TM, and L:TB).

**Statistical analyses.** Statistical analyses were performed using the MetaboAnalyst platform (64, 65). The peak intensity table obtained after preprocessing with MZmine 2 software was uploaded to MetaboAnalyst. The uploaded data file contained 160 (samples) by 3,717 features, including eight groups in the table (leaves, layers of fungus garden [A, B, C, and D] and trash [TT, TM, and TB] material) and deleting 83 features with a constant or single value in all samples. No missing values were found. The interquartile range (IQR) was used to detect filter variables that had nearly constant values. The data were normalized by using a reference feature, in this case, the internal standard sulfamethazine ($C_{12}H_{14}N_4O_2S$) corresponding to *m/z* 279.0910 and with a retention time of 2.55 min. A cube root transformation was performed to facilitate feature comparison (Fig. S2 and S3). Projections to latent structures discriminant analysis (PLS-DA) outputs (coefficient, loadings, scores, and VIP scores) used to generate the corresponding plots shown in Fig. S2 and S3 are available within the MassIVE data set MSV000082636.

**Data availability.** The data sets used in this work were deposited in the online repository GNPS/MassIVE. The data set corresponding to the molecular cartography of *A. texana* fungus garden (MassIVE MSV000082636) can be accessed at https://gnps.ucsd.edu/ProteoSAFe/result.jsp?task=df2eb5792c8446 0b9413fa22af2d0d89&view=advanced_view.

## SUPPLEMENTAL MATERIAL

Supplemental material is available online only.

**FIG S1**, TIF file, 1.6 MB.

**FIG S2**, TIF file, 1.4 MB.

**FIG S3**, TIF file, 1.1 MB.
**FIG S4**, TIF file, 1.8 MB.
**FIG S5**, TIF file, 1 MB.
**FIG S6**, TIF file, 1.8 MB.
**FIG S7**, EPS file, 1 MB.
**FIG S8**, TIF file, 1 MB.
**FIG S9**, TIF file, 1 MB.
**FIG S10**, TIF file, 1.1 MB.

## ACKNOWLEDGMENTS

A.M.C.-R. and P.C.D. were supported by the National Sciences Foundation grant IOS-1656481 and National Institutes of Health (NIH) grant 1DP2GM137413-01. K.E.K., S.P.P., J.L.K., and M.J.B. were supported by NSF grant IOS-1656475. D.P. was supported by the Deutsche Forschungsgemeinschaft (DFG) with grant PE 2600/1. R.D.S. was supported by the São Paulo Research Foundation (awards FAPESP 2017/18922-2 and 2019/05026-4). P.C.D. was supported by the Gordon and Betty Moore Foundation through grant GBMF7622 and the U.S. National Institutes of Health for the Center (P41 GM103484, R03 CA211211, and R01 GM107550). L.-F.N. was supported by the U.S. National Institutes of Health (R01 GM107550). J.J.J.V.D.H. was supported by an ASDI eScience grant, ASDI.2017.030, from the Netherlands eScience Center—NLeSC.

We thank Alan K. Jarmusch and Allegra Aron for their valuable comments in earlier versions of the manuscript and the anonymous reviewers for their suggestions to improve the final version.

P.C.D. and A.M.C.-R. created the concept of applying 3D cartography to fungus gardens. D.P. conceived the idea of chemical proportionality. K.E.K., S.P.P., and A.M.C.-R. deconstructed fungus gardens and prepared extracts. A.M.C.-R. processed samples for LC-MS/MS acquisition. K.E.K. and J.L.K. collected and maintained ant colonies. A.M.C.-R., D.P., R.D.S., M.E., J.J.J.V.D.H., and P.C.D. performed data analysis. L.-F.N. supported FBMN. L.F.N. and A.T. supported molecular formula assignment and structure prediction workflows in the GNPS environment. M.W. supported FBMN and data visualization from GNPS. P.C.D., J.L.K., and M.J.B. provided supervision and funding for the project. A.M.C.-R., J.L.K., and P.C.D. wrote the manuscript. All authors contributed to the writing and editing of the manuscript.

P.C.D. is a scientific advisor to Sirenas. M.W. is Founder of Ometa Labs LLC.

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
