## [Reviewer comments · mSystems]

Chemical gradients of plant substrates in an *Atta texana* fungus garden

Andres Caraballo Rodriguez, Sara Pucket, Kathleen Kyle, Daniel Petras, Ricardo da Silva, Louis-Felix Nothias, Madeleine Ernst, Justin van der Hooft, Anupriya Tripathi, Mingxun Wang, Marcy Balunas, Jonathan Klassen, and Pieter Dorrestein

Corresponding Author(s): Pieter Dorrestein, University of California, San Diego

Review Timeline:

Submission Date:	May 13, 2021
Editorial Decision:	June 9, 2021
Revision Received:	July 1, 2021
Accepted:	July 2, 2021

Editor: Christopher Schadt

Reviewer(s): The reviewers have opted to remain anonymous.

Transaction Report:

DOI: <https://doi.org/10.1128/mSystems.00601-21>

June 9, 2021

Dr. Pieter C. Dorrestein
University of California, San Diego
Department of Pharmacology, Chemistry and Biochemistry
Skaggs School of Pharmacy and Pharmaceutical Sciences
La Jolla, CA 92093

Re: mSystems00601-21 (Chemical gradients of plant substrates in an *Atta texana* fungus garden)

Dear Dr. Pieter C. Dorrestein:

Thank you for submitting your manuscript to mSystems. We have completed our review and I am pleased to inform you that, in principle, we expect to accept it for publication in mSystems. However, acceptance will not be final until you have adequately addressed the reviewer comments.

Thank you for the privilege of reviewing your work. Below you will find instructions from the mSystems editorial office and comments generated during the review. In addition to the comments/requests of the reviewers, please also consider reducing or combining some of the supplemental material. Editorial policy/preference at mSystems is for a maximum of 10 supplemental figures or tables.

Preparing Revision Guidelines

For complete guidelines on revision requirements, please see the Instructions to Authors at <https://msystems.asm.org/sites/default/files/additional-assets/mSys-ITA.pdf>. **Submissions of a paper that does not conform to mSystems guidelines will delay acceptance of your manuscript.**

Corresponding authors may join or renew ASM membership to obtain discounts on publication fees. Need to upgrade your membership level? Please contact Customer Service at

Service@asmusa.org.

Sincerely,

Christopher Schadt

Editor, mSystems

Journals Department
Reviewer comments:

Reviewer #1 (Comments for the Author):

The manuscript "Chemical gradients of plant substrates in an *Atta texana* fungus garden" is an interesting article that presents a chemical perspective of fungus growing ants' symbiotic system originate from plants.

It's about a detailed study based on mass spectrometry that I consider very relevant by applied several metabolomics tools to a unique dataset.

The article structure is adequate, considering the extensive and complex dataset.

I suggest proceeding with normalization of decimal values at m/z and mass error across the manuscript and supplementary material.

Also, I would like to clarify some points:

Line 310: what was the reason to use 2 μM sulfamethazine as an internal standard? Considering its structure and intrinsic chemical ionization that is completely different from the main chemical classes found in the study.

Line 318-319: about the quality control solution applied, it was not clear why the author used this compound mixture. Is this related to mass error correction or retention time monitoring? I found it a little confusing in this regard.

I understand that this article could be used as a guide for other ecological studies that want to apply the metabolomic tools presented.

Reviewer #2 (Comments for the Author):

The manuscript "Chemical gradients of plant substrates in an *Atta texana* fungus garden" provides and very detailed overview of the chemical changes occurring in an *A. texana* fungus garden, in particular demonstrating the capacity of the fungus garden to metabolized plant substrates. This

work was made possible by applying high-throughput analysis (mass spectrometry-based approaches and 3D cartographic analyses) which unable to achieve insights into the chemical transformations found in a laboratory-maintained *A. texana* fungus garden, modifications that were found associated with certain regions of the system.

I will suggest some minor observations:

1. To make a graphical model of the main detected molecules found in the different regions of the system. For example, it is clear that flavonoids and phenolic compounds decrease from top to bottom of the fungus garden, but other compounds such as ergosterol peroxide and shingolipids increased in relative abundance across the layers being enriched in the trash. Additionally, it may be interesting to include some chemical modifications, as those shown for the bioactive molecule ginnalin A.

These detected molecules and transformations could be illustrated in a graphical model that resume the main transformations across the different regions of the system.

2. References. Some journals are abbreviated and some others are not. Others have different format (e.i. PLOS ONE, PlosOne). Also, some scientific names are not in italics.

3. Line 291. Change minutes for min.

4. Line 294. The link for the short video is not available (https://youtu.be/J-Ma_RNj6qs) .

5. In lines 305 to 306 change "Supp Fig S5" and "Supp Fig. S7" for "Supplementary Fig. S5" and "Supplementary Fig. S7".

“Chemical gradients of plant substrates in an *Atta texana* fungus garden”

The manuscript “Chemical gradients of plant substrates in an *Atta texana* fungus garden” provides and very detailed overview of the chemical changes occurring in an *A. texana* fungus garden, in particular demonstrating the capacity of the fungus garden to metabolized plant substrates. This work was made possible by applying high-throughput analysis (mass spectrometry-based approaches and 3D cartographic analyses) which unable to achieve insights into the chemical transformations found in a laboratory-maintained *A. texana* fungus garden, modifications that were found associated with certain regions of the system.

The workflow proposed in this work, opens the opportunity to solve questions related to microbial – ant interactions. Specifically, to identify chemical cues associated with ant behaviors, to determine the effect of plant substrate on the type of chemical transformations and interactions with the fungal garden, to identify the effect of the fungus garden on the microbial communities inhabiting these systems.

Finally, the manuscript is very well written and results are extensively well supported by figures, tables and supplementary data.

I will suggest some minor observations:

1. To make a graphical model of the main detected molecules found in the different regions of the system. For example, it is clear that flavonoids and phenolic compounds decrease from top to bottom of the fungus garden, but other compounds such as ergosterol peroxide and shingolipids increased in relative abundance across the layers being enriched in the trash. Additionally, it may be interesting to include some chemical modifications, as those shown for the bioactive molecule ginnalin A.
These detected molecules and transformations could be illustrated in a graphical model that resume the main transformations across the different regions of the system.
2. References. Some journals are abbreviated and some others are not. Others have different format (e.i. PLOS ONE, PlosOne). Also, some scientific names are not in italics.
3. Line 291. Change minutes for min.
4. Line 294. The link for the short video is not available (https://youtu.be/J-Ma_RNj6qs) .
5. In lines 305 to 306 change “Supp Fig S5” and “Supp Fig. S7” for “Supplementary Fig. S5” and “Supplementary Fig. S7”.

SKAGGS SCHOOL OF PHARMACY AND PHARMACEUTICAL SCIENCES

Mauricio Caraballo, Ph.D.
9500 GILMAN DRIVE
LA JOLLA, CALIFORNIA 92093-0751
Phone: (858) 568-5237
acaraballo@health.ucsd.edu

June 25th 2021

Dear Christopher Schadt
Editor
mSystems

Ref: response to reviewers - manuscript mSystems00601-21

We really appreciate your time as well as the reviewers' to read and comment on our manuscript entitled "*Chemical gradients of plant substrates in an *Atta texana* fungus garden*". We have prepared a revised version of the manuscript based on the provided comments and suggestions. All the changes have been highlighted in the resubmission files. We have also elaborated a point-by-point response to the reviewer's comments shown below in **blue**. Additionally, we modified the supplementary material for a maximum of 10 figures and included their corresponding legends (**Lines 432 - 523**) at the end of the main text as suggested by the Editorial policy at mSystems.

We hope we addressed all the comments and look forward to the Journal's decision.

Sincerely,
Mauricio Caraballo, Ph.D.

Reviewer's Comments (black) and Responses (Blue):**Reviewer #1 (Comments for the Author):**

The manuscript "Chemical gradients of plant substrates in an *Atta texana* fungus garden" is an interesting article that presents a chemical perspective of fungus growing ants' symbiotic system originate from plants.
It's about a detailed study based on mass spectrometry that I consider very relevant by applied several metabolomics tools to a unique dataset.
The article structure is adequate, considering the extensive and complex dataset.

We really appreciate your comments and perception of our work, thanks.

I suggest proceeding with normalization of decimal values at m/z and mass error across the manuscript and supplementary material.

Thanks for the suggestions. For consistency, all the m/z values have been standardized to four decimals, the calc. m/z value and the mass error (ppm) were included for the identified compounds or

when a molecular formula was provided (e.g. Line 91 in the highlighted version of the main text: " $C_{32}H_{52}NO_3$ (calc. m/z 498.3941, error 1.9 ppm) and $C_{30}H_{45}O_3$ (calc. m/z 453.3363, error 4.9 ppm)"). These changes can be found in the highlighted version of the revised manuscript and the supplemental material (legends of supplementary figures, lines 432- 523) found at the end of the main text.

Also, I would like to clarify some points:

Line 310: what was the reason to use 2 μ M sulfamethazine as an internal standard? Considering its structure and intrinsic chemical ionization that is completely different from the main chemical classes found in the study.

Thanks for this question. Yes, the reason we use sulfamethazine as an internal standard, which is also part of the chemical mixture we use as quality control for monitoring LC-MS runs, is part of an effort to be able to cross compare diverse datasets at a global scale. There is not a universal standard protocol in metabolomics, however we use this internal standard to monitor the analyses for retention time and mass shifts throughout runs and batches of the study. Additionally, the internal standard was used for normalizing the data based on peak area of this compound (see Materials and Methods < Statistical analysis), as it's detection is expected to be constant across entire runs under the same analytical conditions within the LC-MS system.

Line 318-319: about the quality control solution applied, it was not clear why the author used this compound mixture. Is this related to mass error correction or retention time monitoring? I found it a little confusing in this regard.

Yes. As mentioned before for the internal standard, the mixture of the six compounds (described in the Materials and Methods section) that are not related to the samples enables us to monitor the LC-MS performance across entire datasets (monitoring retention time and mass shifts) not just from this study. Then, global comparisons at the repository scale and future studies will be possible having these standard protocols.

I understand that this article could be used as a guide for other ecological studies that want to apply the metabolomic tools presented.

Yes, this is correct. The intention of having standard protocols and reproducible workflows are useful for future comparative studies at a larger scale.

Reviewer #2 (Comments for the Author):

The manuscript "Chemical gradients of plant substrates in an *Atta texana* fungus garden" provides and very detailed overview of the chemical changes occurring in an *A. texana* fungus garden, in particular demonstrating the capacity of the fungus garden to metabolized plant substrates. This work was made possible by applying high-throughput analysis (mass spectrometry-based approaches and 3D cartographic analyses) which unable to achieve insights into the chemical transformations found in a laboratory-maintained *A. texana* fungus garden, modifications that were found associated with certain regions of the system.

I will suggest some minor observations:

1. To make a graphical model of the main detected molecules found in the different regions of the system. For example, it is clear that flavonoids and phenolic compounds decrease from top to bottom of the fungus garden, but other compounds such as ergosterol peroxide and shingolipids increased in relative abundance across the layers being enriched in the trash. Additionally, it may be interesting to include some chemical modifications, as those shown for the bioactive molecule ginnalin A.

These detected molecules and transformations could be illustrated in a graphical model that resume the main transformations across the different regions of the system.

This is a great suggestion and we thank the reviewer for this comment. We elaborated **Figure 5**, which is mentioned in the main text (Line 239 "(...) removed from the system as trash (**Fig. 5**)". We also included and labeled additional detected compounds and modifications (compounds 5-11) to better summarize our work as suggested by the reviewer. Figure 3 was also edited to reflect these changes in the text. The manuscript was properly adjusted to these changes and highlighted in the revised version.

Figure 5. Summary of detected molecules and chemical modifications detected in an *Atta texana* fungus garden.

Schematic representation of a laboratory-maintained *Atta texana* fungus garden. From left to right, leaves carried by ants to the fungus garden (middle) and trash accumulated by ants (right). The chemical gradient of plant metabolites showing high abundance at the top layer of the fungus garden is consistent with chemical modifications occurring across the fungus garden. High abundance of fungal metabolites associated with the trash indicates that these are chemical labels of the discarded material removed by the ants. Some examples of detected molecules and chemical modifications are shown. Chemical structures of trash-associated metabolites remain to be elucidated. Color codes of the compounds are as indicated at the top of the figure. This image was partially created using BioRender.com

2. References. Some journals are abbreviated and some others are not. Others have different format (e.i. PLOS ONE, PlosOne). Also, some scientific names are not in italics.

Thanks. We have adjusted the references for the manuscript and the supplementary material according to the format used by the mSystems journal.

3. Line 291. Change minutes for min.

Changed (Line 284 in the highlighted document).

4. Line 284. The link for the short video is not available (https://youtu.be/J-Ma_RNj6qs) .

We corrected this, as the link provided in legend for Figure 1 (https://youtu.be/_ikhKelfrY8) is the correct one which is also public now, as well as in Line 287 under the Material and Methods section. Thanks for pointing this out.

5. In lines 305 to 306 change "Supp Fig S5" and "Supp Fig. S7" for "Supplementary Fig. S5" and "Supplementary Fig. S7".

Corrected. Lines 299-300 now read "kaempferol (**Supplementary Fig. S4**)" and "quercetin (**9, Supplementary Fig. S5**)", respectively.

July 2, 2021

Dr. Pieter C. Dorrestein
University of California, San Diego
Department of Pharmacology, Chemistry and Biochemistry
Skaggs School of Pharmacy and Pharmaceutical Sciences
La Jolla, CA 92093

Re: mSystems00601-21R1 (Chemical gradients of plant substrates in an *Atta texana* fungus garden)

Dear Dr. Pieter C. Dorrestein:

Your manuscript has been accepted, and I am forwarding it to the ASM Journals Department for publication. For your reference, ASM Journals' address is given below. Before it can be scheduled for publication, your manuscript will be checked by the mSystems senior production editor, Ellie Ghatineh, to make sure that all elements meet the technical requirements for publication. She will contact you if anything needs to be revised before copyediting and production can begin. Otherwise, you will be notified when your proofs are ready to be viewed.

As an open-access publication, mSystems receives no financial support from paid subscriptions and depends on authors' prompt payment of publication fees as soon as their articles are accepted. =

Publication Fees:

We recognize that the video files can become quite large, and so to avoid quality loss ASM suggests sending the video file via <https://www.wetransfer.com/>. When you have a final version of the video and the still ready to share, please send it to Ellie Ghatineh at eghatineh@asmusa.org.

Sincerely,

Christopher Schadt
Editor, mSystems

Journals Department
Fig. S9: Accept
Fig. S6: Accept
Fig. S8: Accept
Fig. S1: Accept
Fig. S2: Accept
Fig. S4: Accept
Fig. S7: Accept
Fig. S10: Accept
Fig. S3: Accept
Fig. S5: Accept